# Automated Chlorine Dosage in a Simulated Drinking Water Treatment Plant: A Real Case Study

**Javier Gámiz [1,2] , Antoni Grau [1,*] , Herminio Martínez [3] and Yolanda Bolea [1]**

1   Automatic Control Department, Technical University of Catalonia, 08034 Barcelona, Spain;
    jgamiz@agbar.es (J.G.); yolanda.bolea@upc.edu (Y.B.)
2   Industrial Control Systems Department, Aigües de Barcelona, 08038 Barcelona, Spain
3   Department of Electronics Engineering, Technical University of Catalonia, 08034 Barcelona, Spain;
    herminio.martinez@upc.edu
*   Correspondence: antoni.grau@upc.edu



**Featured Application: This research was applied in the Sant Joan Despí drinking water treatment plant in Barcelona city, providing drinking water to 1,900,000 inhabitants. The tool was validated with real data obtained in the plant in different conditions of effluents.**

**Abstract:** In this paper, we present a simulator of a drinking water treatment plant. The model of the plant was based in hydraulic and matter transportation models. In order to not introduce more inaccuracies in the simulation, the control system was based in the real equipment deployed in the plant. This fact was the challenging part of the simulator, and an accurate design is presented in this research, wherein the sampling time had to be limited to interchange data between the SCADA in the plant and the simulator in real time. Due to the impossibility to stop the plant when testing the new control strategy, a simulator implemented the plant behavior under different extreme conditions. The validation of the simulator was performed with real data obtained from the plant.

**Keywords:** drinking water treatment plant; advection–diffusion–reaction; chlorine dosage; simulator validation; industrial application

---

## 1. Introduction

Drinking water is a necessary but scarce good. Despite the fact that water is the most abundant and common substance on our planet—as it covers 70% of its surface—96.5% of it is contained in the oceans. Of the remaining 3.5%, approximately 2.5% is found in the polar ice caps and glaciers and only 0.61% is liquid fresh water. Of the latter, around 0.98% is found in underground aquifers, which are difficult to access, while only 0.009% constitutes fresh surface water (rivers and lakes). Furthermore, only 0.003% of the total is fresh water available to be used for residential purposes. That is, if the Earth's total water were a 100 L container, only half a teaspoon of water would be suitable for human consumption [1]. The sources of water pollution can be natural (rain, decomposing vegetable matter, soil erosion, etc.) or anthropogenic (activity livestock, by-products of industrial activity, home waters, etc.), but both give rise to water that does not meet the necessary requirements to ensure its potability. The basic water treatment processes include several stages: coagulation, flocculation, particle separation (sedimentation/flotation), filtration, and disinfection (chlorination/ozonation) [2]. In many of these stages, the addition of chemicals into the flow of water to be treated is performed, and it is at this very point that this research provides a contribution with respect to correct dosage and control. Of all the treatments mentioned above, this paper focuses on disinfection. This process

attempts to destroy or inactivate pathogenic organisms present in water, mainly bacteria, viruses, and protozoa [3].

Disinfection treatments can be physical (gamma radiation, X-rays, ultraviolet radiation, thermal sterilization, etc.) or chemical (heavy metals, acids or bases, halogens, ozone, permanganate, etc.), with the latter being the most common form of treatment. Among the chemical reagents, chlorine and its derived compounds are the most widely used disinfecting agents worldwide. Many organisms regulate the residual chlorine values and they depend on the end use of the water. Thus, for drinking water, it is recommended that the residual free chlorine be between 0.2 and 1 ppm (parts per million), while in the case of swimming pools and spas, it should be kept between 1.5–3.0 ppm. However, these values are general and each competent body has determined its own thresholds.

The use of chlorine as a disinfecting agent began in the early 20th century and went on to complete the filtration process, which was already widely used. The most common chlorine family products for water disinfection are chlorine gas, chloramines, sodium hypochlorite, and calcium hypochlorite. Chlorine ($Cl_2$) is a yellowish green, denser than air, toxic gas. It is a very oxidizing product that reacts with many compounds [4]. In fact, the most widely used chemical in the world is chlorine.

In the presence of humidity, chlorine is extremely corrosive and therefore the conduits and the materials in contact with it must be made of special alloys. Chlorine vapor is irritating by inhalation and can cause serious injury if exposed to high concentrations. Chlorine management must therefore be carried out by specialized staff, and very effective control and alarm systems are necessary. For these reasons, the use of hypochlorites in solution or in solid form is preferable. Sodium hypochlorite (NaClO) in solution is a disinfectant that has been used since the 18th century and is popularly known as bleach. At an industrial level, it is obtained by reacting chlorine gas with a sodium hydroxide solution. After the reaction, greenish yellow aqueous solutions are obtained, which have a determined concentration of active chlorine per liter [5]. It is marketed in solutions with concentrations between 3% and 15% by weight. Sodium hypochlorite is a very powerful and unstable oxidant, and thus a solution of 100 g of active chlorine per liter, after being stored for 3 months, can contain 90 g or even less. Calcium hypochlorite ($Ca(ClO)_2$) is a white solid with a content of between 20% and 70% active chlorine. It is highly corrosive and can ignite on contact with certain acidic materials. However, it has two advantages over sodium hypochlorite: its higher chlorine content and its greater stability. To be used, it is diluted with water to obtain a more manageable concentration solution, for example 2%. When $Cl_2$ dissolves in water, it rapidly hydrolyses to generate hypochlorous acid and hydrochloric acid.

$$Cl_2 + H_2O \quad \leftrightarrow \quad HClO + HCl$$

In the case of hypochlorites, the dissociation of both salts occurs according to the following equations:

$$NaClO + H_2O \quad \leftrightarrow \quad NaOH + HClO$$
$$Ca(ClO)_2 + 2H_2O \quad \leftrightarrow \quad Ca(OH)_2 + 2HClO$$

Therefore, hypochlorous acid, which is actually the disinfectant variety, ends up forming in either case chlorine, sodium hypochlorite and calcium hypochlorite.One of the disadvantages of using chlorine and its derivatives is that it reacts with a large amount of organic matter and gives rise to trihalomethanes (THM), many of which have been shown to be toxic or carcinogenic [6]. Another drawback is the formation of chlorophenols in waters containing phenols, which would lead to bad odors. Chlorine also reacts with ammonia dissolved in water to form chloramines. These products also have some disinfecting power, although they are approximately 25 times less effective than free chlorine. However, their residence time in the water is long and, for this reason, they have sometimes been used as a reserve for residual chlorine. They present two major drawbacks: they can give rise to odors and flavors and are potentially toxic in a chronic way. Figure 1 shows an interesting representation of the risks associated with disinfection with chlorine. We adapted this figure from [7], indicating the optimal point for chlorination; due to the dynamics of chemical and microbiological risks, the optimal point of chlorination is at the intersection, in order to avoid both risks. It is clear that

not dosing can lead to a very high risk of infection (microbiological risk), but in any case, overdosing is not a valid solution, as it does not guarantee the elimination of health risks, since it favors the increase of chemical risks. An insufficient dose can cause vomiting or diarrhea (microbiological risk) and an overdose has carcinogenic effects (chemical risk); thus, the objective is an optimal dose. At very high levels of chlorine, the microbial risk increases, as taste and odor may cause the use of unsafe supplies.

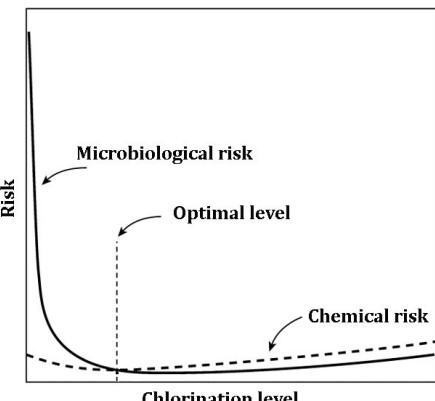

**Figure 1.** Relationship between chlorination level and associated risk, adapted from [7].

On the basis of the foregoing, it can be deduced that chlorine (and derivatives), in addition to reacting with microorganisms, also reacts with other matter dissolved in the medium: organic matter, metals (iron, manganese), and mainly plastic derivatives. For this reason, to achieve a certain level of residual chlorine, the necessary amount to be added is much higher than the residual obtained. Therefore, before deciding on the dose of chlorine to be used to disinfect, the demand for chlorine must be determined, that is, the amount of chlorine consumed within the tank until the residual appears at the end of the plant, prior to entering the water distribution step.

Regarding the legislation of drinking water, there are some regulations at different levels. The higher level in Europe is dictated by the European Parliament and of the Council in its Directive 2008/105/EC on environmental quality standards in the field of water policy [8]. This legislation guarantees that water intended for human consumption can be consumed safely and without danger to health. Specifically, in Spain (valid in the current drinking water treatment plant (DWTP) location), the regulation is articulated through royal decrees RD 140/2003, RD 314/2016, and RD 902/2018 [9], specifying that the residual chlorine value must be between 0.2 and 1.0 ppm at all points in the supply network. In short, this legislation will be replaced by the future European Directive on Drinking Waters, which is at the moment in the final revision phase, with the date of possible approval at the end of 2020 and with mandatory entry in 2022. This new standard requires a series of new requirements and is mainly aimed at avoiding unnecessary water loss and helping to reduce the carbon footprint of the EU member states. The new water directive emerges to achieve the 2030 sustainable development goals (Goal 6) and the Paris agreement on climate change. Information will be key to increasing consumer confidence in tap water, and thus information on the quality and supply of drinking water in each area should be provided on the Internet. The idea is that the more information there is, the more confidence there will be about this resource that comes out of our taps, and therefore there will be a lower purchase of plastic bottled water, reducing the waste from this material.

The main goal of this research was to develop a simulator of a drinking water treatment plant (DWTP). New technologies deployed in this kind of plant allow for the automatization of the chlorination process, and the use of this novel simulator will allow for the testing of any chlorination strategy without involving such a critical infrastructure. The plant cannot stop delivering drinking water to a large portion of the population and the tests cannot be done with the plant at full capacity. The simulator will have the ability to be connected in real-time with automatic chlorination system

(specific hardware in the plant), and all this hardware equipment does not need to be simulated, thus enhancing the reliability of the whole process.

## 2. Materials and Methods

### 2.1. Simulator Design

This is one of the most important contributions of the presented research. The design of the simulator starts with defining the requirement, and proposing a novel architecture that will be able to fulfill them. This proposal involves the software implementation of different mathematical models (hydraulic and transportation models) and a hardware interconnection between plant computers, that is, programmable logic controller (PLC) and the main computer executing the simulator.

The chlorination process in the plant has been subjected to different phases of improvement at the automation level in recent years. The problems inherent in the automatic control have been diverse: variability in the chemistry of chlorine, punctual existence in the water to deal with significant amounts of ammonium, large dead times due to the hydraulics of its facilities, or disturbances caused by sudden changes in flow. The use of a simulator that reflects in the most realistic way the behavior of the chlorination process greatly facilitates the design of the control system in the effluent and allows decisions to be made on the optimization of the actual installation. Simulation has been seen over time as a modelling tool that has a very broad development and does not require sophisticated mathematics or statistics to develop a model and its usage [10]. On the other hand, the realization of experiments or tests on the real plant was expensive because of the need to have additional electronics and infrastructure, and moreover it mainly is especially dangerous because this is the last process of the DWTP and there is no correction mechanism in case of error (overdosage or underdosage of chlorine in tap water). The DWTP cannot be closed for testing new chlorination strategies, and the large duration of tests cannot be afforded by all the citizens and small industries that need drinking water uninterruptedly. Therefore, we decided to use a simulator that would allow for the study and analysis of the most appropriate online controller to later transfer the simulation results to the real control system of the plant.

When selecting a simulator to implement the desired model, we defined four strong requirements due to the complexity of the plant and the implications of stopping it:

1.  The simulation tool has to be able to simulate a hydraulic model considering the turbulent flow rate.
2.  The tool has to simulate a model of transport of diluted species where the reaction of chlorine with compounds of different sources would take place.
3.  The tool needs a connection to the plant control system in real time through the OPC (object linking and embedding (OLE) for process control) platform that allows testing.
4.  The tool has to allow importing Supervisory Control and Data Acquisition (SCADA) data and be able to compare it, offline, with those produced by the model implemented in the simulator.

In recent years, there is a growing interest by large automation companies to integrate computational fluid dynamics (CFD) software into online or real-time control scenarios. Through analyzing different software or platforms existing in the market to be used as a simulator in the chlorine automatic dosage, we finally decided to implement our own simulator, mainly for the simplicity of equations that we developed in this research. With our proposal, it is possible to reduce the sampling time to near 1 second. This fact is very important in order to have a good description and knowledge of water behavior and chlorine diffusion into tanks. After accurate research among many commercial simulators (18 software packages in total), we did not find any commercial simulator fulfilling the aforementioned conditions with such a reduced sampling time that was open-source and internally customized. Next, the architecture of the proposed simulator is presented, which would allow testing chlorination control algorithms for treatment plants (DWTP), fulfilling the following features:

- Large contact tanks.
- Variability in the demand for chlorine by water quality.
- Considerable disturbances due to the appearance of ammonium.
- Disturbances in the water inflow.

One of the main advantages of working with a model implemented in an open source language is the possibility to start with a simple approximation of the process and then gradually refine (programming) the model as the understanding of the process improves. The continuous process of refining leads to achieving a good (accurate enough) approximation to solve the problem of the chlorination tank simulator fulfilling the previously mentioned requirements.

The contribution of this paper is the design and implementation of such a simulator. The architecture depicted in Figure 2 is the key of the contribution, and the starting point for the implementation. With the proposal of this architecture, we will fulfil the four requirements described above. The simulator integrates the equations to describe a hydraulic model and a transport model of diluted species—in this case, the chorine reaction.

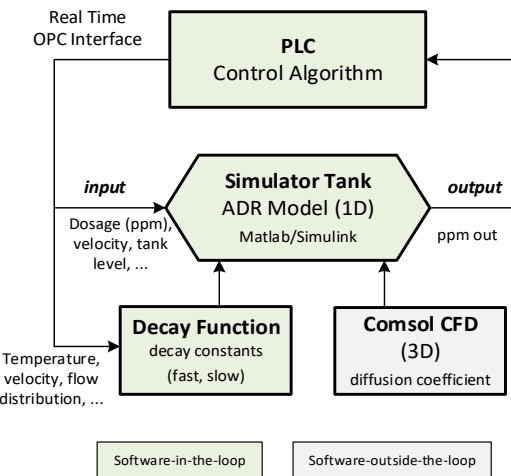

**Figure 2.** General architecture of the simulation system for tank chlorination.

The tank behavior (*Simulator Tank* block) was implemented by means of a Simulink S-function, which works in real time, communicating the simulator with the control algorithm implemented in the PLC. The communication of the control loop is the OPC platform. This communication will allow for the importing of SCADA real data from the plant to compare and validate the simulator behavior. In addition to the control variable, the output of the controller transfers the water speed and the level of the tank to the simulator. Conversely, the simulator provides two outputs: (1) the response of the simulated plant to the input of the control algorithm, and (2) the error between the measured chlorine dose and the necessary dose. On the other hand, the output of the controller provides a set of variables such as temperature, speed, or the distribution of flows between osmosis and carbon filters, among others, which allow the chlorine decay to be estimated using the *Decay Function* block. The *Comsol CFD (3D)* block provides a table of diffusion coefficients for locations at different distances of the chlorine output meter of the treated tank in an offline process. In the next sections, we present how this design was implemented and validated, showing relevant results compared with real data.

### 2.2. Description of Sant Joan Despí DWTP

Since its inception in 1955, the DWTP of Sant Joan Despí (Barcelona, Spain) has aimed to purify the surface waters captured from the Llobregat River and groundwater from the delta aquifer of the river, today supplying drinking water to more than half of the inhabitants (1,900,000) of the city of Barcelona, as well as the southwest conurbation.

The DWTP in Sant Joan Despí incorporates the latest technology in drinking water treatment. The schematic of the plant is shown in Figure 3. Since the water is captured in the Llobregat river, the following processes are applied: surface water capture and desanding; peroxidation; pumping water to the decantation tanks (tag 5 in Figure 3); filtration by sand; pumping by means of four Archimedes screws, wherein the water is raised so that it can follow the process by its own gravity; ozonation; filtration by activated carbon filters; and reverse osmosis, to finally arrive at the mixing and chlorination chambers (tags 13, 14 in Figure 3). In these chambers, the water from the two treatment lines is mixed and chlorination takes place, which ensures the removal of almost all the ammonium content remaining in the water. The study of this paper concentrated mainly on these tanks. Finally, two pumping stations inject water to the distribution network at different levels to be supplied.

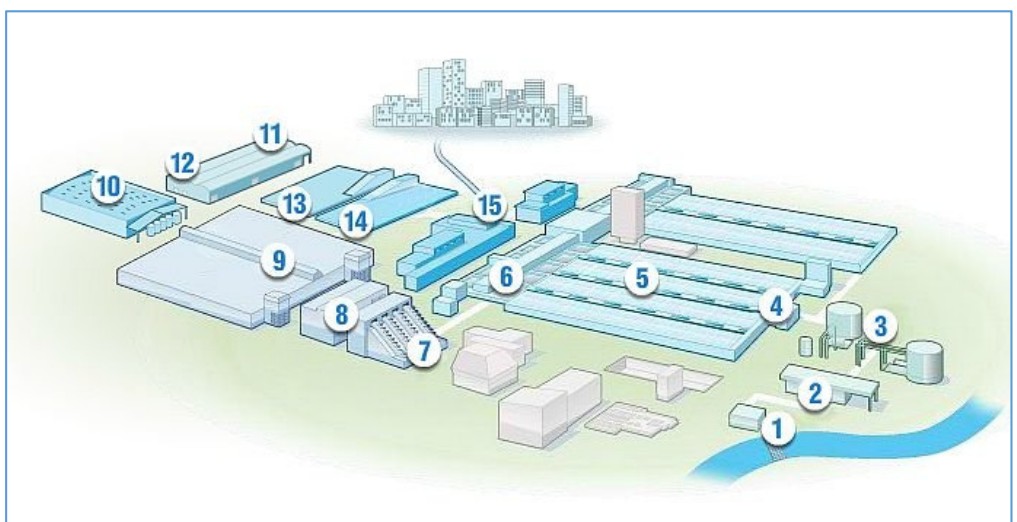

**Figure 3.** Drinking water treatment plant (DWTP) at Sant Joan Despí schematic.

The DWTP has a series of processes prior to chlorination in the effluent that favor, to a greater or lesser extent, the elimination of organic matter of natural source and those derived from human activities.

One of the last processes of the plant is chlorination, before driving the treated water to the tanks distributed throughout the city of Barcelona. In [11], there is a good explanation of how the Barcelona water network is implemented, not only physically, but also with an extensive set of sensors to obtain chlorine concentration at the storage tanks distributed along the city, with a total amount of 200 sensors for a distribution network of 4600 km. These tanks, close to the final destination of drinking water (domestic homes), have sensors that measure the lack of chlorine during its transportation from the DWTP to the storage tanks, and a last rechlorination is done to ensure a concentration in the range of 0.8 to 1 ppm when reaching the taps.

To better understand the power of the DWTP, Table 1 contains a set of parameters of water taken at different points in the plant, specifically, the sampling points located: (i) at the entrance of the plant, the intake; (ii) at the entrance of the mixing chamber in the chlorination tanks; and (iii) at the output of chlorination tanks when the drinking water is ready to be transported and distributed through the water network. The chlorination process was performed appropriately because the ammonium concentration dropped to 0 ppm. This is one of the most critical parameters to control to avoid, and the rest of parameters were expected to have those values after all the processes involved in the plant.

In the following sections, the chlorine disinfection strategy in both tanks of the plant is described. Before the process was carried out automatically by the control and instrumentation electronics and the supervision system, we performed the process manually. The facilities and equipment that compose the chlorine regulation system are detailed as well. It should be borne in mind, however, that the validation step described in this article was carried out in the first tank (tank 1) where the ammonium

appears and can be detected. The same process would be totally valid when applied to the second tank (tank 2).

**Table 1.** Parameters of water at different sampling points in the DWTP.

| Parameter (Unities) | Llobregat River Intake | Mixing Chamber * | Chlorination Tank Outlet ** |
|---|---|---|---|
| Turbidity (NTU) | 100 | 0.2 | 0.2 |
| Ammonium concentration (ppm) | 0.25 | 0.02 | 0 |
| Chlorine (ppm) | 0 | 2 *** | 1.2 |
| pH (0–14) | 7.8 | 7.3 | 7.3 |
| Temperature (°C) | 20 | 20 | 20 |
| Conductivity (µS/cm) | 1200 | 900 | 900 |

\* Chlorination tank inlet; ** to distribution network; *** this value is due to dosage at the mixing chamber.

### 2.3. Disinfection Strategy

Under normal operation, the water previously treated in the different processes of organic matter elimination enters the disinfection phase by combining the diluted chlorine gas and the water to be treated in two large tanks. The retention time required for disinfection is a delay time that makes feedback control a difficult task. The disinfection strategy is based on covering the basic demand of chlorine in the first tank, that is, the needs for chlorine demand are met as a result of the consumption of certain existing organic and inorganic matter. Then, in the second tank, the necessary chlorine is added, that is, free chlorine, to cover losses (by chemical reactions, not leakages) by its transportation. In the same way, the added chlorine needs to reach a certain level to comply with the legal regulations in the effluent (see references above for regulations), with a maximum value of 1.0 mg/L for a good taste and a minimum of 0.2 mg/L to ensure the death of pathogens, bacteria, and viruses. Currently the concentration values required at the exit of the DWTP are usually between 0.8 and 1.2 mg/L of free chlorine.

As shown in Figure 4, the dosing stages to be carried out in the first tank (tank 1) would be between zones 1 and 3. The water to be treated would come into contact with the chlorine in the first tank (zone 1) to react quickly with inorganic matter such as $Fe^{++}$, $Mn^{++}$, $H_2S$, or organic matter. Once the initial demand for chlorine has been satisfied as dosing continues, the creation of monochloroamines, dichloroamines, and trichloroamines (zone 2) begins due to the combination of free chlorine with ammonium or its derivatives. In drinking water, monochloroamines are deliberately formed by the reaction of chlorine in an aqueous solution with the added ammonium ion or non-dissociated ammonia in water [12]. In zone 2, total chlorine is a combination of free chlorine and its different organic compounds, and its disinfection effectiveness is limited when total chlorine is only formed by free chlorine. If the dosage continues increasing (zone 3) when all the nitrogen-derived species have oxidized, the reaction reaches a breakpoint of special importance in the chlorination process because beyond this breakpoint the entire dosage will be fully effective in terms of disinfection.

From the beginning of zone 4, all the dosed chlorine becomes residual free chlorine, becoming the ideal point of entry to the tank 2. Specifically, the disinfection strategy that is followed in the DWTP is to raise the outlet from tank 1 to 0.2 ppm to ensure that tank 2 operates only with free chlorine.

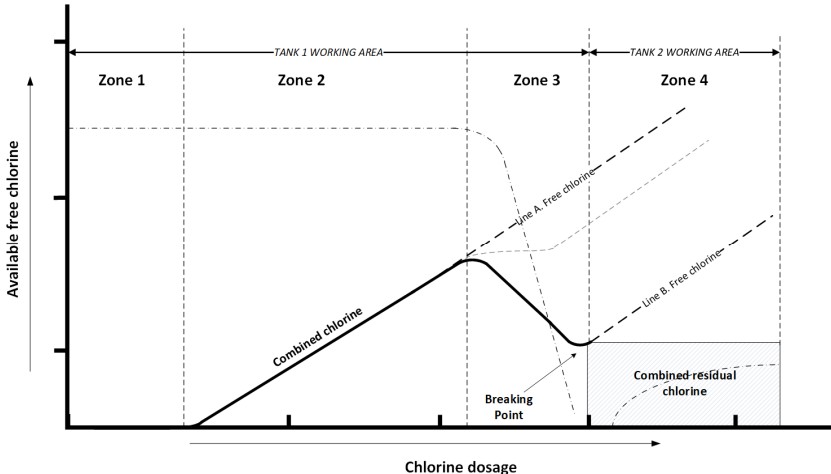

**Figure 4.** Transition in the dosing areas of the chlorination in tank 1 and inlet of tank 2.

### 2.4. Tanks and Equipment

As previously mentioned, the disinfection system has two contact tanks, tank 1 and 2, with a capacity of 10,000 m³ each. These tanks are hydraulically coupled and, in case of an anomaly in any of them, the DWTP can treat water only with the available tank (only for a limited time of operation). The water flow to be treated in tank 1 (Figure 5) arrives at a small chamber where water treated with a traditional treatment by granulated active carbon filters (GACF) is mixed with water treated with a reverse osmosis (RO) treatment. At the exit of both treatment processes, there are flow meters that allow sensing the flow coming from both the osmosis and the carbon filters. The sum of these two flows will be the variable of total flow to be treated, as well as the input for the control system and the simulator presented in this article. The actual flow range treated by the plant is between 5.5 m³/s at maximum performance and a minimum flow in certain operating situations of 1.0 m³/s. In normal regime, the distribution by GACF and by RO is 50%, however, this distribution may change in a few hours due to the operating conditions of the DWTP. The theoretical residence time of water in the one tank can vary between 23 min and 55 s for a flow rate of 5.5 m³/s up to 2 h and 11 min for a flow rate of 1.0 m³/s.

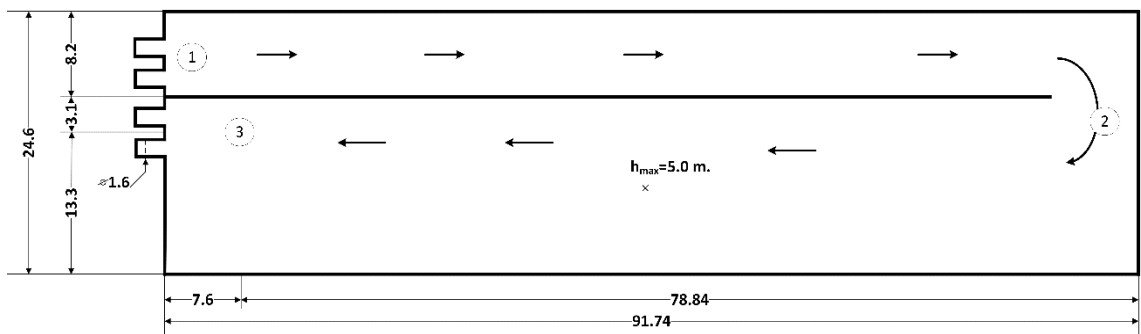

**Figure 5.** Plant view of chlorination tank 1 (measured in meters).

In the inlet of tank 1, the mixing chamber, we installed an ammonium meter (An1), allowing the control system to acquire the amount of ammonium contained in the water to be treated, with the first chlorinated water dosage point (1) being located a few meters apart downstream of the entrance.

It is essential for the control system to know the delay times between the production of chlorinated water and the dosage in point 1 since they are dead times that must be compensated and considered in the calculation of the dose. At a distance of 87 m from the dosing point 1, there is the first residual chlorine meter (2). This distance between the dosing point and the measurement of free chlorine is

considered sufficient in most cases for the chlorine to react with organic matter and other compounds. Depending on the velocity of the water and the amount of ammonium, sometimes the measurement is no longer representative for errors in the meter and for this reason there is a second meter (3) located at the exit of the tank. As shown in Figure 5, the tank is divided into two parts and forces the treated water to travel twice the distance so that the reaction is complete and the contact time is sufficient for an effective disinfection. In the center of the tank, there is an ultrasonic level meter with a height range of 0–4 m.

The analytical instrumentation provided by the control system is composed of a colorimetric ammonium-type analyzer at the entrance of the tank and two amperometric meters that measure the residual chlorine at the entrance and exit of the two tanks. The response of the amiometric residual chlorine meters for combined chlorine samples has been quite satisfactory once the reaction time of the ammonium with the free chlorine is guaranteed. To measure the flow at the output of each of the 20 GACF and the 10 RO frames, we used electromagnetic flow meters. It is important to parameterize these flowmeters at the level of integration of the measure to dampen possible sudden changes in the reading of the control system. Finally, an ultrasonic level meter is available in both tanks that allow us to know the variations of the volume stored in each tank on the basis of the knowledge of their dimensions.

## 3. Mathematical Process Modeling

In this section, the graybox model implemented is presented. This model allows for a simulation of the chlorination process that describes the behavior of the fluid by the advection and diffusion equations, as well as a chlorine-demand estimation function to model the reaction.

The chlorine dosing system in contact tanks 1 and 2 can be characterized as a distributed parameter system that takes place in both coupled reactors. The chlorine dosed at the beginning of each tank (reactor) is subject to a first-order decay which allows the mass balance shown in Equation (1). This partial differential equation (PDE) represents the model of a solute in a turbulent flow, in 3D:

$$\frac{\partial c}{\partial t} = D_{t-x}\frac{\partial^2 c}{\partial x^2} + D_{t-y}\frac{\partial^2 c}{\partial y^2} + D_{t-z}\frac{\partial^2 c}{\partial z^2} - U\frac{\partial c}{\partial x} - V\frac{\partial c}{\partial y} - W\frac{\partial c}{\partial z} \tag{1}$$

where $c$ is the chlorine concentration; $D_{t-x,y,z}$ is the dispersion; and $U$, $V$, and $W$ are the water velocities in the coordinate axes $x$, $y$, and $z$, respectively. Simplifying it to a 1D equation and adding the reactive term, we obtained the following equation:

$$\frac{\partial c}{\partial t} = D\frac{\partial^2 c}{\partial x^2} - U\frac{\partial c}{\partial x} - kc \tag{2}$$

where $U$ is the water velocity in the tank (m/s). Therefore, Equation (2) is the advection–diffusion–reaction equation (ADR) and belongs to the group of partial differential equations of parabolic type.

The simulator developed and presented in this article is based on the discretization of Equation (2) [13], treating the chlorine decay part ($kc$). The diffusion coefficient values were obtained from the Comsol CFD Software on the basis of the tank and flow characteristics, treated as a RANS $k$-$\varepsilon$ turbulence model.

The proposed simulator, despite working in 1D, collects all the necessary information for the characterization of the process and allows the control system to be tested as if it were actually being dosed in the tank. In the following sections, we detail the more relevant aspects and considerations that were considered at the time of the design, as well as the implementation of the simulator related to advection transport, flow characteristics and the parameterization of the chlorine reaction, and diffusion of the chlorine through the tank.

In the following sections, the turbulent flow regime (Reynolds number) is verified, and the Péclet number is estimated in order to assess the influence of the convective versus diffusive effect. Finally, the Schmidt number chosen for this turbulent flow model is presented.

### 3.1. Flow Characteristics

The flow in the tank under study corresponds to a three-dimensional speed distribution. This is a non-stationary flow, considering the fluctuation of the flow in the tank and not its uniformity due to its geometry considering the characteristic velocity profile of the flow in open channels.

The consideration of constant velocity throughout the domain would be a relatively correct simplification if the chlorine were evenly distributed at all times. However, in principle, this hypothesis is not acceptable due to the characteristic velocity profile for this type of geometry. In addition, the geometry of the studied domain should be considered where changes occur in the direction of flow. This situation even drives away the actual behavior of the fluid from the idealized one-dimensional flow situation.

On the other hand, we faced a turbulent flow, according to the Reynolds number range, Equation (3), for the featured width of the tank $L_{(width)}$ = 8.2 m and according to the properties of the water at normal working temperature, with a velocity for maximum and minimum operations points [0.052 . . . 0.128]m/s being

$$Re = \frac{\rho v L}{\mu} = \frac{1000 \times v \times 8.2}{0.001} \tag{3}$$

where $\rho$ is the density of the fluid (gr/L), $v$ the velocity, $L$ the tank width, and $\mu$ the dynamic viscosity of the fluid. The water velocity range was obtained on the basis of the flow and the elapsed time to cross the tank of known dimensions. The minimum and maximum values for the Reynolds number were $4.29 \times 10^5$ and $1.06 \times 10^6$, respectively, defining a range of values within the turbulent regime. In [14], a comparative study is presented between different configurations of contact tanks for a disinfection process and the calculation of the Reynolds number, using as a parameter, in this case, the kinematic viscosity of the water as opposed to the dynamic viscosity such as in this research.

### 3.2. Diffusion Coefficient

The diffusion coefficient depends largely on the nature of the particles, the solvent, the temperature, and the viscosity of the solvent. In this case, neither the type of particles nor the solvent is modified during the process. On the other hand, temperature variations are not important with respect to diffusion and, as a consequence, there are also no significant variations in the viscosity of the solution. Considering all these conditions, we can affirm that the diffusion coefficient will remain constant.

In general, if there is an active flow, as presented in this paper, the diffusion effect is negligible. However, to determine the relationship between the advective and diffusive terms, the Péclet number $Pe$, Equation (4), was estimated:

$$Pe = \frac{L \times v}{D} \tag{4}$$

where $L$ is the distance between concentration measurement points, $v$ the fluid velocity, and $D$ (m/s$^2$) the diffusion coefficient.

In this study case, the Comsol CFD software was used to obtain the value of the diffusion coefficient of the particles from the characterization of the tank and different flow rates giving a range of 0.0055 to 0.013 m$^2$/s for water flow in the range between 1.5 m$^3$/s and 3.7 m$^3$/s, respectively. Similar to this strategy, in [15], CFD software was used to obtain the dissemination number of the treated tank.

For the calculation of the minimum and maximum values of the number of Péclet, according to the extreme values of the velocity, we took the most unfavorable value of the diffusion coefficient ($D$ = 0.013 m$^2$/s). Considering these conditions, the Péclet number oscillated between 71.92 and 1831.0, both higher than 1. These values suggest that the diffusive term was negligible compared to the effects

of advection, and since the diffusion can be neglected for *Pe* >> 1 under these conditions, the effects of advection exceeded those of diffusion in determining the overall mass flow.

### 3.3. Turbulence Model

The turbulence model chosen for the CFD study is the R-based k-epsilon (*k-ε*) model, which is a two-equation model that provides a general description of the turbulence through turbulent kinetic energy (*k*) and dissipation (*ε*). The *k-ε* turbulence model is widely used to model flow behavior in chlorination tanks of these characteristics [16].

The Schmidt number (*Sc*), Equation (5), is a relevant parameter in the configuration of the model used in this study, which is used in RANS *k-ε* to avoid the resolution of the boundary layer. The *k-ε* model predicts turbulent viscosity thanks to the Schmidt number:

$$S_c = \frac{\mu}{\rho D_t} \tag{5}$$

where $\mu$ is the dynamic viscosity of the fluid, $\rho$ the density of the fluid, and $D_t$ is mass diffusivity of the fluid—a value that depends on the fluid, water plus chlorine in this case.

The Schmidt number is an empirical constant with typical values between 0.1 and 1; in the present study giving a value of 0.7, following the criteria of many other documented works on water treatment and contact tanks of similar characteristics [17–20]. This parameter is relatively insensitive to the properties of the molecular fluid (the particular value obtained from other experiments can be used despite the differences between the simulation domain). It represents a significant parameter for fully developed turbulent flows, and it is considered that, for the present case, the average Reynolds number is in the transition to the turbulence zone with moderate values, and thus it was not a key parameter for the present case. A sensitivity analysis of the *Sc* was carried out in the present study, showing the little relevance of the changes in its value close to 0.7; between 0.5 and 0.9 the concentration variations obtained in the effluent were around ±0.01 ppm. Finally, with respect to *Sc*, it should be borne in mind that the experimental determination of this parameter was not within the scope of this project.

### 3.4. Chlorine Reaction through the Tank

The expression in Equation (2) is a generalization that must be developed with more precision for the present case. In [21,22], different models are presented, describing a chlorine decay more adjusted to the reality of the studied plant. Considering the characteristics of the water that reaches the exit of the DWTP, a significant part of the chlorine (fraction *f* in Equation (6)) reacts quickly with the existing matter (organic and inorganic) and the rest (fraction 1–*f*) of the matter reacts with the remaining chlorine fraction. Therefore, a combined first-order model according to [21], plus a combination of first and second order presented in [22], yields Equation (6):

$$\frac{\partial c}{\partial t} = D \frac{\partial^2 c}{\partial x^2} - U \frac{\partial c}{\partial x} - k_R c(f) - k_r c^2 (1-f) - k_s c(1-f) \tag{6}$$

where $k_R$ is the decay coefficient for the rapid reaction, and $k_r$ and $k_s$ would be the decay constants of rapid and slow reaction for the remaining chlorine fraction with different characteristics to those that react following $k_R$. Let *f* be the fraction of chlorine that reacts quickly in response to the decay constant $k_R$.

It has been proven by laboratory analysis that, for treated waters without ammonia, the demand for chlorine varies between 0.2 ppm and 0.5 ppm depending on whether its source is from the traditional treatment of granular active carbon filters (GACF) or osmosis (RO). These reactions take place between 5 to 10 min after the chlorine is exposed to the water and follows a decay according to the constant $k_R$. The demand for chlorine is very high, and $k_R$ varies significantly when the existence of ammonium occurs, with values between 7 and 10 times the amount of $NH_4$-N detected by the ammonium

meter (An1). Constants $k_r$ and $k_s$ were estimated in [23], with values between $3.3934 \times 10^{-5}$ (s$^{-1}$) and $5.4259 \times 10^{-7}$ (s$^{-1}$), respectively. These values ($k_r$ and $k_s$) are significant because of the importance in the parameterization of the proposed simulator. With those values of $k_r$ and $k_s$, and knowing that the demand for chlorine in its rapid reaction stage is between 0.2 ppm and 1.0 ppm and that the fraction that reacts with the chlorine is around 40%, then $k_R$ is between $3.046 \times 10^{-4}$ (s$^{-1}$) y $14.05 \times 10^{-4}$ (s$^{-1}$), according to experimental data.

Figure 6 shows the simulator response to a step in the dosage of 1 ppm for a flow of 1.5 m$^3$/s parameterized with $k_r$ and $k_s$, and different values of $k_R$ with a reaction fraction of $f = 0.4$ (40%).

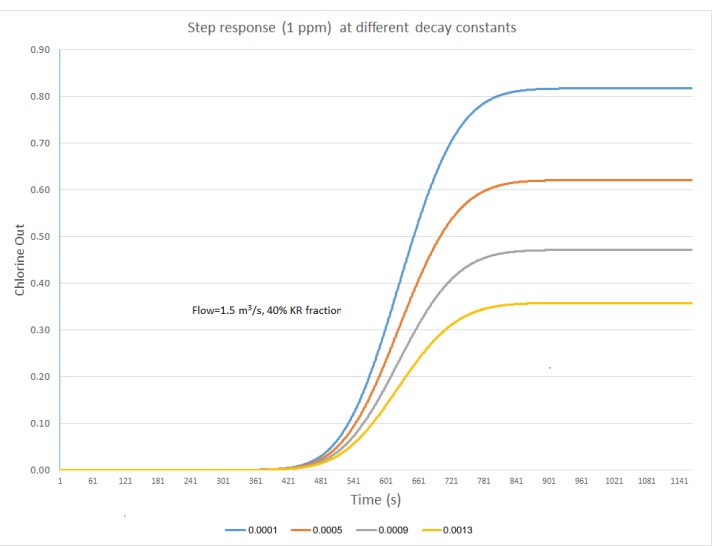

**Figure 6.** Step response for a dose of 1 ppm of chlorine for different values of $k_R$ (0.0001, 0.0005, 0.0009, 0.0013 s$^{-1}$).

Figure 7 shows a view of one of the studies carried out in steady state with the Comsol CFD on the diffusion of chlorine along the contact tank. Specifically, on the basis of a color scale, the diffusion of chlorine along the tank is represented for a turbulent flow rate RANS $k$-$\varepsilon$ with a flow rate of 2.7 m$^3$/s, with a decay constant of $9.0 \times 10{-4}$ (s$^{-1}$), a Schmidt number of 0.7, and an initial dosing step of 2 ppm of chlorine.

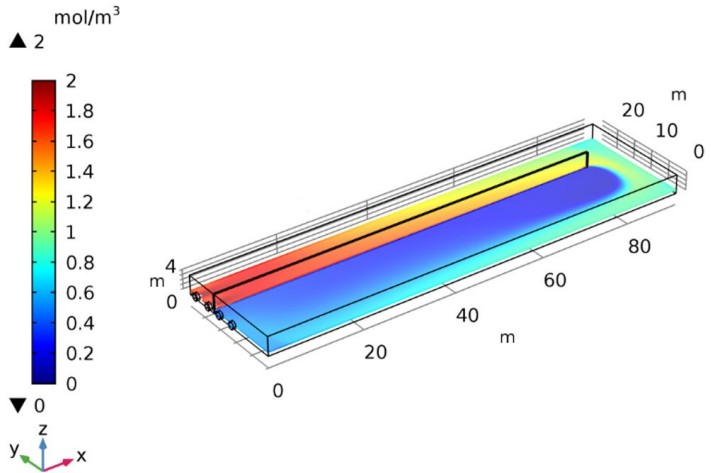

**Figure 7.** Comsol simulation for decay constant of $k_R = 9.0 \times 10^{-4}$ (s$^{-1}$).

## 4. Discretization

The partial derivative equation (PDE) presented in Equation (6) in continuous space was reduced to a second-order ordinary differential equation (ODE), Equation (7), in the discretization of the

simulator. It was considered that the type of simulator flow belongs to the "Plug Flow" pattern where the fluid circulates through the tank evenly and along parallel paths from the entrance to the exit of the tank.

$$0 = D\frac{d^2C}{dx^2} - U\frac{dC}{dx} - k_R c(f) - k_r c^2(1-f) - k_s c(1-f) \tag{7}$$

Among the different methods of discretization for the type of equations that govern the behavior of the simulated system process (Quick, Upwind, etc.), we chose a central finite differences equation, which was adapted in an acceptable way for the values of flow, diffusive characteristics, and Péclet numbers of the present problem [24]. In [25], different discretization methods were presented and discussed to address advection, diffusion, and reaction problems in contact tanks; in [26], a discretization scheme was presented for storage tanks in transport processes and distribution of the same type of water treated by the DWTP of this article.

When applying central finite differences by replacing the first and second derivative in Equation (7), then Equation (8) is obtained:

$$0 = D\frac{C_{i+1} - 2C_i + C_{i-1}}{x^2} - U\frac{C_{i+1} - C_{i-1}}{2x} - k_R c_i(f) - k_r c_i^2(1-f) - k_s c_i(1-f) \tag{8}$$

for a diffusion coefficient D and a chlorine concentration c at different instants of time.

Figure 8 shows a diagram of the diffusion effect in chlorine concentration. It supposes a tank of length *L* with constant increments of space on the mesh ($\Delta$x) at different periods of time. An increment of $\Delta$x = L/(n − 1) was considered, with *n* being the number of mesh divisions. To avoid the effects of dynamic instability, the increment values of x fulfilling $x \leq \frac{2D}{v}$ were restricted, and the stability criterium $t \leq \frac{(x)^2}{2D + k_R x^2}$ was applied according to [27]. Applying the stability restriction on the spacing $\Delta$x in the most unfavourable case, a value of $\Delta$x ≤ 0.2 m was obtained and $\Delta$t ≤ 2.16 s. In the simulation, knowing that the actual dosage analyzer 2 (An2) is located at 83 m from the tank 1 entrance, a value $\Delta$x = 0.127 m and $\Delta$t = 1 s with a mesh of *n* = 685 slots was used, in accordance with a trade-off to minimize the error with a computing time that would allow the simulation in real time.

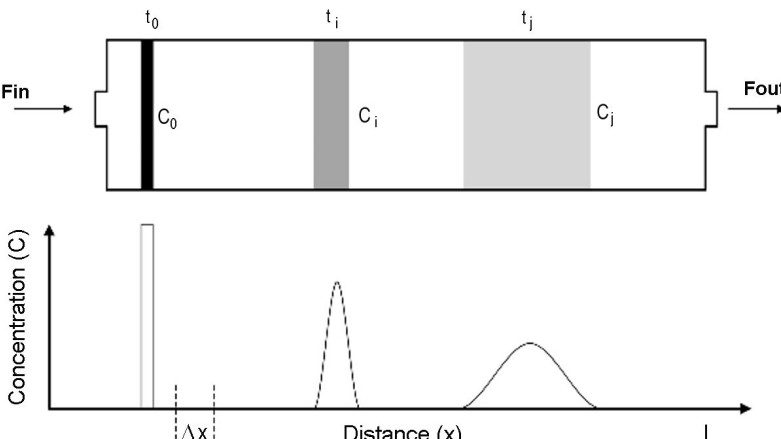

**Figure 8.** Diagram representing the diffusion effect respect time and space (not at scale) in the tank for a water inflow *Fin*. At different instant times ($t_0$, $t_i$, $t_j$ ... ), the concentration of chlorine is different in the tank ($c_0$, $c_i$, $c_j$ ... ) depending on the distance to the tank inlet (function of *x*).

## 5. Simulator Implementation

For the implementation of the simulator, the Matlab/Simulink tool was used using the C language to encode the functions that describe the behavior of the model. The simulation code was programmed and encapsulated in an S-Function (Level 2) in Simulink environment and the interface was adapted to be able to contrast data from the output of the supervision and control software (SCADA) with the output of the simulation. The proposed simulator could be configured for any programmable logic

controller (PLC) and SCADA controller with an OPC interface (OLE for process control, object linking, and embedding). One of the advantages of using a S-Function is that it allows for the creation of a general-purpose block that can be used in all the iterations of the model, varying the parameters with each instance of the block and thus interacting in real time with the controller (PLC). The simulator implements the central finite differences for the discretized model in Equation (8) [28].

Simulation main screen (in Simulink environment) for the interaction scheme between the simulator and the PLC controller according to the simulator architecture (Figure 2) is shown in Figure 9. The data from the controller are evaluated by the *estimateCd* function that calculates the chlorine demand and, thus, the value of the rapid decay coefficient ($k_R$) for each iteration. Depending on the velocity of the flow, we determined the diffusion coefficients that are part of the input variables of the simulator, together with the dose, $k_R$, and velocity. The designed interface allows for the setting of the distance to the reading point, the rapid reaction fraction for $k_R$, and constant values $k_r$ and $k_s$.

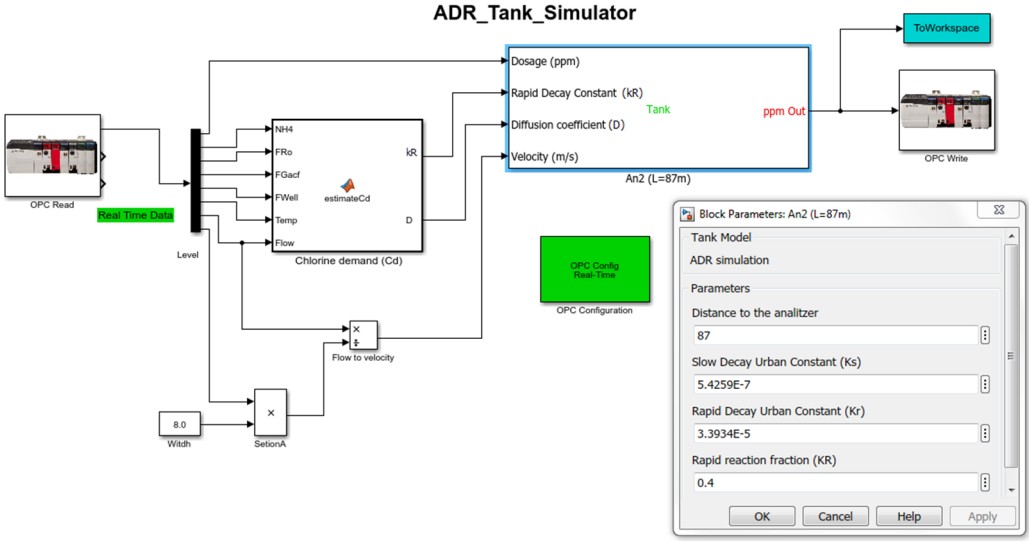

**Figure 9.** Connection between the simulator and a programmable logic controller (PLC) in the OPC (object linking and embedding (OLE) for process control) platform.

The simulator comprises two working zones. If there is less than 0.02 ppm of ammonium in the treated water (sensed with analyzer An1), the chlorine demand is calculated on the basis of the temperature and the source of treated water. We developed a classification of the chlorine demand on the basis of the water temperature (Temp) and the percentage of water passing through the carbon filter treatment (TPGacf) with respect to the total treated flow. In the tree, each child node represents a chlorine demand value based on the classification by temperature and source and percentage of treated water. This study was generated through taking real data from the last 5 years in the plant. Figure 10 (left) shows the decision tree for classified data. The study was implemented with RStudio. For values greater than 0.02 ppm in the ammonium concentration, the meter yields accurate and reliable results, having proven experimentally that there is a linear correlation (greater than 0.93) between the ammonium concentration and the chlorine demand for treated water (see Figure 10, right).

The validation of the simulator is an important part of this work, because it allows for the measurement of the goodness of such a simulator in its use in forecasting different situations in the real plant in front unknown events that can happen in the future. Results are shown in the next section, but here the implementation of how the simulation is validated is presented.

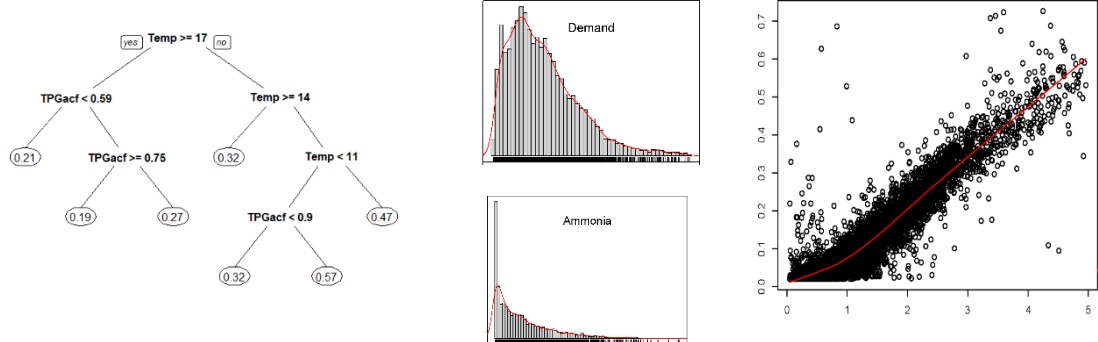

**Figure 10. Left**: decision tree for chlorine demand; **right**: correlation between ammonium concentration and chlorine demand (about *R* = 0.93).

The procedure to compare real data obtained with the SCADA and simulated data is shown in Figure 11. Two blocks (*Tank_secA* and *Tank_secB*) are instantiated in the test, one for each installed analyzer, An2 and An3, at 87 and 163 m away from the dosing point, respectively. The level of the tank (*Level*) and the flow rate (*TotalFlow*) are used to calculate the velocities in sections A and B of the tank. The function blocks (*An2 Sample* and *An3 Sample*) simulate the measurement behavior of chlorine analyzers with periodic samples every 5 min. The *Chlorine Dosage System* function block simulates the process of injecting chlorine gas into water from its generation point and its delay in transport to the contact tank, in order to be more realistic in the simulation procedure.

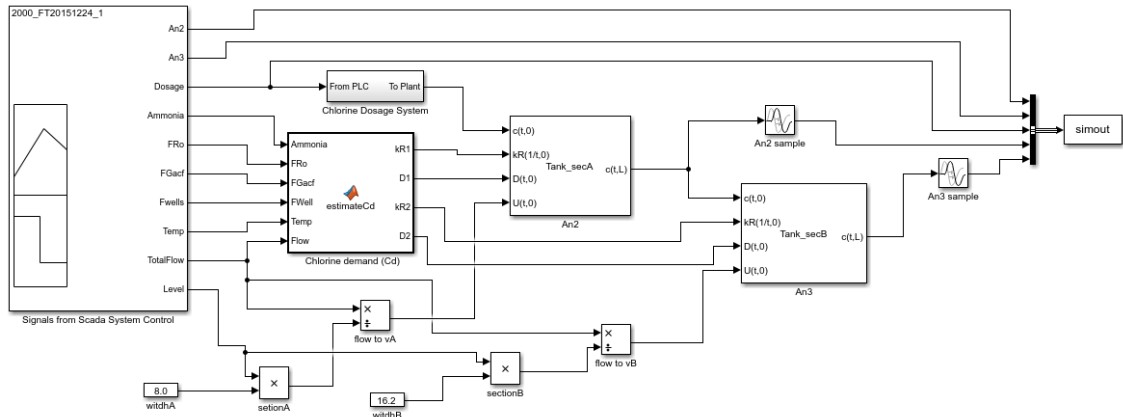

**Figure 11.** Simulator block diagram to compare simulation data and real data from SCADA.

At present, the chlorination stage is performed in a semi-automatic regime. From the plant control room, the operators check the level of chlorine concentration on the basis of the sensors' readings, and they order PLC (programmable logic controllers) to open the valves of chlorine bottles, which adds a specific dosage to reach the demand at the control point (located as indicated in Figure 5).

## 6. Results: Validation of the Simulator

The validation of the simulator was divided into two stages with specific objectives. In a first stage, data from the Comsol CFD, considered as a reference computational fluid dynamics software, were contrasted with respect to the output of the simulator for conditions of advective flow, diffusion, and decay of chlorine. The objective of this phase was to verify the correct implementation and validation of the advection–diffusion–reaction model in the Simulink S-Function. In a second stage, the *estimateCD* function was parameterized and implemented for the characteristics of the water treated by the DWTP, and data from the SCADA database were validated on the basis of this parameterization.

Table 2 shows a comparison of values obtained from Comsol CFD with respect to those provided by the simulator for different decay constants and a flow rate of 3.7 m³/s. In any case, the relative error

in the sample is larger than 2% considering that chlorine meters have an absolute error of 0.04ppm. This fact implies that the error made by the simulator would widely meet the requirements in terms of accuracy.

**Table 2.** Comparison of chlorine decay between the simulator and the Comsol computational fluid dynamics (CFD).

| $k_R$ (s$^{-1}$) | Chlorine Concentration (ppm) for a Flow Rate of 3.7 m$^3$/s | | | | | |
| | At 35 m from Tank 1 Inlet | | | At 70 m from Tank 1 Inlet | | |
| | Comsol CFD | Proposed Simulator | Relative Error at 35 m | Comsol CFD | Proposed Simulator | Relative Error at 70 m |
|---|---|---|---|---|---|---|
| 0.0001 | 0.9180 | 0.9028 | −1.69% | 0.8869 | 0.8802 | −0.76% |
| 0.0003 | 0.8620 | 0.8554 | −0.77% | 0.7859 | 0.7859 | −0.01% |
| 0.0005 | 0.8107 | 0.8105 | −0.02% | 0.6978 | 0.7016 | 0.55% |
| 0.0007 | 0.7634 | 0.7680 | 0.60% | 0.6206 | 0.6264 | 0.93% |
| 0.0009 | 0.7198 | 0.7277 | 1.09% | 0.5529 | 0.5593 | 1.15% |
| 0.0011 | 0.6794 | 0.6895 | 1.48% | 0.4934 | 0.4994 | 1.21% |
| 0.0013 | 0.6419 | 0.6534 | 1.76% | 0.4409 | 0.4459 | 1.13% |
| 0.0015 | 0.6071 | 0.6192 | 1.95% | 0.3946 | 0.3982 | 0.91% |

Figure 12 shows graphically a comparison of data obtained with Comsol CFD and the simulator, with different water flows between 1.50 and 3.70 m$^3$/s. On the basis of these results, the simulator gave excellent results.

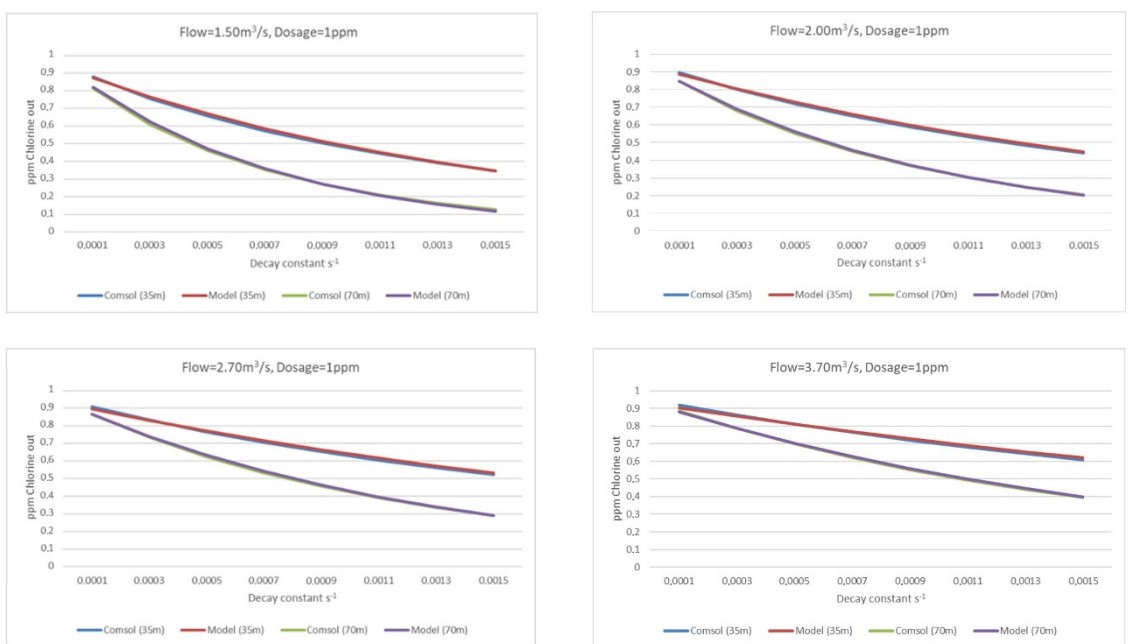

**Figure 12.** Results comparison between CFD Comsol data and simulator data for different flow rates (1.50, 2.00, 2.70, and 3.70 m$^3$/s).

However, the most important validation of simulator was carried out with real data; this is perhaps the most effective way to demonstrate the power of a simulator. In this case, the experiments were performed with real data obtained from the plant in real situations. Most of the time, the plant has neither variation on the input flow nor the effluent where the water comes from, and thus the chlorine dosage is constant. Such situations are easy to model. However, when weather conditions are varying, such as in episodes of flooding or big storms, or even in large episodes of draught when the river flow in very low, the situation in the plant becomes exceptional. In such periods, the chlorine dosage has to abruptly change due the bad conditions of input flow to the plant. We focused our attention upon such

days, and obviously to the associated data recorder during those extreme episodes. Together with the flow, the dosage is already recorded as in a log file. Therefore, it is easy to reproduce the conditions that generated a specific behavior in the plant, and those conditions were used as input in the simulator to recreate the real situation that occurred in the plant. The output of the simulator was compared with the chlorine concentration in the measurement point (output of the plant) as a consequence of the dosage that was injected in the inflow water. This comparison study was carried out in all the operating points of the plant, that is, at the different input flows of 1.5, 2.0, 2.7, and 3.7 m³/s.

As can be seen in Figure 13, the plant behavior refers to the chlorine concentration at the end of the tank, that is, the concentration of chlorine in parts per million ready to enter the pipes for distribution. The model is the execution result of the simulator as the chlorine concentration at the same point of the plant, and therefore plots in Figure 13 compare both concentration in the same location of the tank. As it can be observed, the simulator behavior worked exceptionally well following the reality in front of different input flow situations. This qualitative assessment aside, the variations between real data and simulated data are quantified in Figure 14.

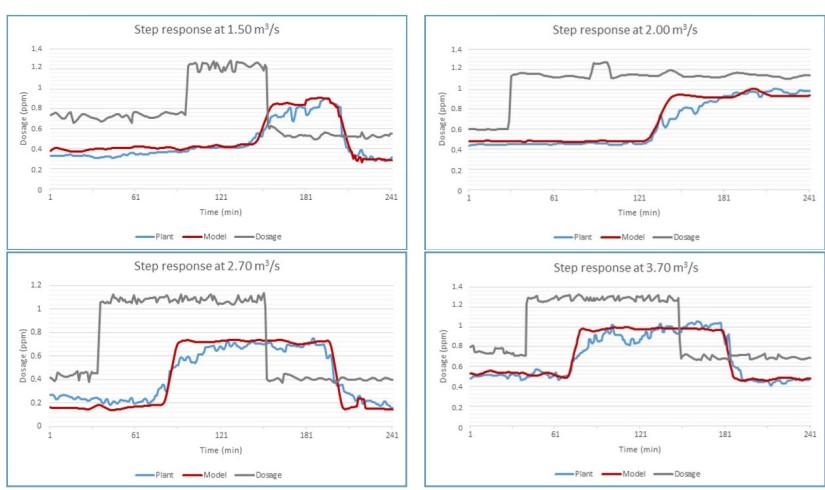

**Figure 13.** Data comparison between real data from SCADA and simulator data for different flow rates (1.5, 2.0, 2.7, and 3.7 m³/s).

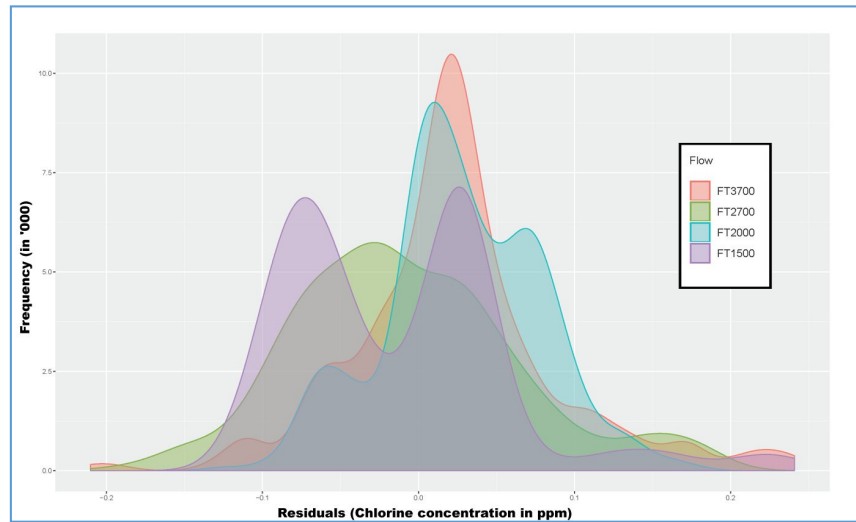

**Figure 14.** Plot of residuals for different flow rate operating points (1.5, 2.0, 2.7, and 3.7 m³/s).

Defining a residual value (or residuals with sign) as the difference between the observed values and the values predicted by the model (estimated values), Figure 14 shows a graph of the residual values obtained in the different operating points. For all flow values, a distribution of bell-shaped

residuals can be seen, except for flows close to 1.5 m$^3$/s, which is multimodal. At large flow rates with high velocity (above from 1.5 m$^3$/s), it is easier to predict errors and residuals closer to zero and with less dispersion, because the dispersion, either multimodal or unimodal, increases when the flow drops. One explanation is that the simulation treats the problem as a case of one-dimensional flow (and Plug Flow), and it does not consider that the velocity is not uniform throughout the domain. Not all the dosed chlorine moves at the same velocity in the tanks and it is also possible that there are fluid recirculations in the studied domain. As a consequence, at the measurement point, the concentration is detected with a time lag with respect to the simulation, and at low flow rates this fact is greatly accentuated. Finally, the chlorine analyzer error of ± 5% over 5.00 ppm must be considered and, therefore, those simulation errors can be perfectly accepted. The abscissa axis indicates the error in concentration of chlorine between real and simulated data. The ordinate axis represents the frequency of such residuals appears, that is, a kind of histogram of number of appearances. Note that the frequency is in thousands because we studied large amounts of data, searching for exceptional situations in the plant, and then for every particular study the sequence is for 241 min, sampled at 1 s, the sampling time of the designed simulator.

As it can be seen in Table 3, the mean square error (MSE) are collected as a measure of the quality of the simulator. In this case, the difference between the real data and the simulated data is measured as:

$$MSE = \frac{1}{N} \sum_{t=1}^{N} \left( c_{real\_data} - c_{simulated\_data} \right)^2 \tag{9}$$

that is, the sum of the square differences between the chlorine concentration in the real plant and the result of the simulator in front the same dosage and flow. It is worthwhile to note that variable *t*, the time, has a step of 1 s.

**Table 3.** Mean square error (MSE) between the observed values (real data) and the estimated values (simulated data).

| Operating Point (Input Flow) | MSE Error |
| --- | --- |
| 1.5 m$^3$/s | 0.071307 |
| 2.0 m$^3$/s | 0.057033 |
| 2.7 m$^3$/s | 0.072119 |
| 3.7 m$^3$/s | 0.070618 |

## 7. Conclusions

In this article, we proposed the design and implementation of a simulator that allows for the verification of control methodologies in a simple and visual way before its implementation directly in a real drinking water treatment plant DWTP. The proposed simulator is specific for processes that involve a hydraulic model where transport of diluted species can be simulated. The simulator provides simplicity, easy connection to plant control equipment (with OPC platforms), and reliability. The connection between the specific hardware of the plant and the simulator operated in a satisfactory manner, allowing a data interchange at a sample rate of 1 s, time enough in this kind of phenomenon. The simulator was validated with a more complex simulator (Comsol CFD) unable to operate with the plant equipment. The result of such a comparison was very satisfactory, giving an error less than 2% in the worst case. Moreover, and a crucial test, the simulator was validated with real data acquired under extreme circumstances in tough periods in terms of water effluents arriving to the DWTP. The results were completely satisfactory, with a good performance. Those good results confirmed that the reduction of the model from partial derivatives equations to ordinary differential equations was correctly performed, and thus the model is reliable enough to be confident in its functioning in the plant. After these results, we are confident in the use of the simulator as a forecast tool in a real plant.

**Author Contributions:** Conceptualization, J.G. and A.G.; methodology, J.G. and Y.B.; software, J.G. and H.M.; validation, H.M. and Y.B.; formal analysis, A.G. and H.M.; investigation, J.G. and Y.B.; resources, A.G. and H.M.; data curation, J.G. and Y.B.; writing—original draft preparation, J.G.; writing—review and editing, A.G.; visualization, H.M. and Y.B.; supervision, A.G. All authors have read and agreed to the published version of the manuscript.

**Acknowledgments:** We would like to thank Francisco Luque Montilla; David Ibarra from Aquated, Inc.; Aigües de Barcelona analytic laboratory staff; Ricardo Torres, UPC; and Mercedes Garcia, Education Consortium Barcelona.

**Conflicts of Interest:** The authors declare no conflict of interest.

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
