# Peer review of "Automated Chlorine Dosage in a Simulated Drinking Water Treatment Plant: A Real Case Study"

_applsci, doi:10.3390/app10114035_

Round 1

Reviewer 1 Report

It is important theme, and the point is clear, so I judged this paper worth publishing.

Author Response

Authors would like to thankful this reviewer for considering that our research is under important subject. Hope that the revised paper would please this reviewer as well. 

Reviewer 2 Report

Journal: Applied Science

Title: Automated chlorine dosage in a drinking water treatment plant. A real case study

The authors investigate the possibility of automated chlorination in a drinking water treatment plant in Barcelona by a mathematical simulation procedure described in detail in Chapter 5, based on five years of real collected data obtained from drinking water treatment plant.

The manuscript is well structured in all its parts. I would say that chapters 2 and 5 are written in too much detail, especially chapter 2 which is quite well known, but I do not diminish their quality. While reading the manuscript, I had some questions that I found answered by reading the manuscript.

My comments follow:

I miss the word "simulation" in the title of the manuscript.

The manuscript should state how chlorination is carried out in the said plant, it is obviously not automated. With this you would emphasize the importance of your research.

Your research was based on the results of residual chlorine measured at the end of the tank. I think it is extremely important to have the results of residual chlorine at greater distances from the plant, for example at the most remote points of water distribution. Usually with increasing distance the residual chlorine concentration decreases.

Author Response

Authors would like to thank this reviewer for his/her valuable comments and suggestions. Attached you will find a letter responding to the review. 

Reviewer 3 Report

The research results presented by the authors in “Automated chlorine dosage in a drinking water treatment plant. A real case study” are interesting, however not clear enough. The paper includes too much overall information and its structure is not suitable for scientific publications. It is difficult to separate the theory from the authors' achievements. General information should be provided only in the introduction. I  have several comments, which would make it more understandable.

Abstract

Too much overall information. The aim of the research should be clearly stated here. There is lack of conclusions driven from the research.

Introduction

This part is too general and unrelated to the topic of the article. I encourage you to remove it completely and start the Introduction with the information about disinfection given in the Background section. Please, have a look:

Lines 32-39 – lack of the reference;

Line 42-54 – unnecessary and not clear - relation between the indicator 6.4.2? and water stress. How the water stress is measured? What does the % in the legend of figure 1 mean?

Lines 58- 69 – too general and trivial for such as article,

Line 71 -  “home waters” what is that?

Lines 79-84 – should be removed.

At the end of introduction part, state clearly the aim of the research.

Background => move to Introduction

Data presented in table 1 concerns different disinfectants. They don't contribute anything, so I suggest you remove them.

Line 136 - Can you give some information about information taken from [8]?

Line 146 – “...” What do you mean using …? Just explain it. It is better to avoid such signs in scientific papers.

Materials and methods – lack of such a subtitle

Simulator Design = > should be a subtitle of: Materials and methods

This is very important part of your paper. However it should be redrafted to highlight the design of the simulator

Chlorination Plant  => should be a subtitle of: Materials and methods

 “Chlorination plant” – better “Description of Sant Joan Despi DWTP”.

Line 296 “Figures 6 and 7 depict some of areas in the DWTP explained by the plant schematic.”  Figures could be removed, because you have given a wide characteristics of the plant in lines 245-285. However, I suggest to redraft this paragraph and leave only information on the chloration plant. Remove the description of the whole water treatment plant and other processes.

Please provide short information about chemical composition of water (intake and water before mixing and chlorination chambers).

Results – clearly indicate this part in the article.

Conclusions

A few more conclusions were made on the basis of the research. They are given somewhere in the paper, but you should repeat them in here.

Author Response

(The authors gave the same response as above.)

Round 2

Reviewer 2 Report

The authors improved the manuscript according to my suggestions and explained everything clearly in the answer, so my decision is acceptance of the manuscript in this form.

Reviewer 3 Report

Thank you for taking my comments into account.

Please make a small correction to the introduction part - line 35. (rain, decomposing vegetable matter, soil erosion...) => (e.g.rain, decomposing vegetable matter, soil erosion). The same in lines: 36, 46. Correct double spaces.